# Improving Prenatal Diagnosis Precision for Congenital Clubfoot by Using Three-Dimensional Ultrasonography

**DOI:** 10.3390/diagnostics14010117

**Published:** 2024-01-04

**Authors:** Yoo-min Kim, Ji Su Seong, Ji Hoi Kim, Na Mi Lee, In Ho Choi, Yejin Jo, Gina Nam, Gwang Jun Kim

**Affiliations:** 1Department of Obstetrics and Gynecology, Chung-Ang University Gwang-Myung Hospital, Chung-Ang University College of Medicine, Gwangmyung 14353, Gyeonggi-do, Republic of Korea; shygirl1227@naver.com (Y.-m.K.); jsseong0323@gmail.com (J.S.S.); fathoi@naver.com (J.H.K.); 2Department of Pediatrics, Chung-Ang University Gwang-Myung Hospital, Chung-Ang University College of Medicine, Gwangmyung 14353, Gyeonggi-do, Republic of Korea; piena81@cauhs.or.kr; 3Division of Pediatric Orthopedics, Chung-Ang University Hospital, Chung-Ang University College of Medicine, Seoul 06973, Republic of Korea; inhoc1@cauhs.or.kr; 4Department of Mathematics, Chung-Ang University, Seoul 06974, Republic of Korea; cindyye@naver.com; 5Department of Obstetrics and Gynecology, Chung-Ang University Hospital, Chung-Ang University College of Medicine, Seoul 06973, Republic of Korea; ginanam@cau.ac.kr

**Keywords:** clubfoot, congenital abnormalities, three-dimensional ultrasound, foot malpositions, prenatal diagnosis, ultrasound

## Abstract

Prenatal diagnosis of clubfoot traditionally relied on two-dimensional ultrasonography. To enhance diagnosis and predict postnatal outcomes, we examined the parameters that differentiate pathological clubfoot using three-dimensional ultrasonography. In our retrospective study, we examined the findings of prenatal ultrasound and the postnatal outcomes of pregnancies with suspected congenital clubfoot between 2018 and 2021. Based on the three-dimensional perspective, we measured the angles of varus, equinus, calcaneopedal block, and forefoot adduction and compared the sonographic variables between the postnatal treated and non-treated groups. We evaluated 31 pregnancies (47 feet) with suspected clubfoot using three-dimensional ultrasonography. After delivery, a total of 37 feet (78.7%) underwent treatment involving serial casting only or additional Achilles tenotomy. The treated group showed significantly greater hindfoot varus deviation (60.5° vs. 46.6°, *p* = 0.026) and calcaneopedal block deviation (65.6° vs. 26.6°, *p* < 0.05) compared to the non-treated group. The calcaneopedal block had an area under the curve of 0.98 with a diagnostic threshold of 46.2 degrees (sensitivity of 97%, specificity of 90%, positive predictive value of 97%, and negative predictive value of 90%). During prenatal evaluation of clubfoot using three-dimensional ultrasonography, the calcaneopedal block deviation has the potential to predict postnatal treatment.

## 1. Introduction

Two-dimensional (2D) ultrasonography (US) has been widely used as a prenatal diagnostic tool for fetuses with suspected clubfoot. For decades, recent advances in fetal imaging have improved the accuracy of prenatal diagnosis of clubfoot, achieving a success rate of about 80% [1]. However, the false positive rate (FPR) still varies from 10% to 25% [2,3,4,5,6]. One significant factor contributing to the inability to reduce the false positive rate is the resemblance in features between pathologic and benign feet observed during 2D US examinations. Classically, prenatal clubfoot is diagnosed based on the fact that the long bones of the tibia and fibula are characterized in the coronal plane as part of the lateral or plantar aspect of the foot, with the supinated and adducted positions. 

Although the sonographic findings of pathologic clubfoot were similar to those of benign clubfoot, the treatment strategies and outcomes differed between the two groups. Pathological clubfoot is a subluxation of the talo-calcaneo-navicular joint with stiffness of the midtarsal and ankle joints, which causes underdevelopment of the calf and peroneal muscles and requires postnatal reductive treatment [7]. In the 1980s and 1990s, surgical correction was the primary treatment. In the long-term follow-up analysis of the surgical correction, no difference was found in the functional plantigrade foot compared to that of normal adults, but limitations in movement due to pain and premature arthritis were observed in adults with surgical correction [8]. Currently, the gold standard treatment of pathologic clubfoot is the conservative Ponseti method, a non-invasive treatment for clubfoot, with a high success rate of 90–100% and reduction of deformity with weekly serial plaster casting and Percutaneous Achilles tenotomy [9]. Open surgery is the treatment of choice when conservative treatment is unsuccessful in the correction of the deformity [10,11]. In contrast, benign clubfoot can be easily corrected with gentle positioning without any postnatal treatment. Therefore, the two variants of the clubfoot spectrum need to be accurately differentiated during the prenatal period to allow appropriate counseling of parents and adequate planning of neonatal management. 

After delivery, decisions related to the type of treatment depend on the Demiglio scoring system conducted by experienced orthopedic surgeons, which is a typical scoring method with high intra- and inter-observer reliability [12,13,14]. The four major components, equinus, varus, derotation of the calcaneal block, and forefoot adduction, are assessed by manual examination and are used to evaluate reducibility. Postnatal evaluation of repositioning ability is based on physical examination by multiple approaches. However, prenatal measurements using 2D US only evaluate the angle between the foot and ankle in a single plane, limiting the ability to predict treatment outcomes. Previous studies focused only on evaluations in the coronal plane using 2D US and did not consider a comprehensive approach [2,5,15,16]. 

Recently, research results have been reported suggesting that the simple prenatal evaluation method using 2D ultrasound should be developed into a more detailed evaluation method by using 3D US [17,18]. Marie et al. suggested that three rigorous approaches are important for differentiating between pathological and benign clubfoot to improve precise prenatal diagnosis [17]. Lanna et al. found that measuring the angle of the lower leg and ankle among the clubfoot assessment methods through a multi-angle approach that measures the thickness of the calf muscle, the degree of curvature in the plantar plane of the foot, and the supination angle of the ankle could reduce the false positive rate in predicting postnatal treatment methods [18]. 

As pathologic clubfoot involves complex subluxation, including equinus, varus, calcaneopedal block, and adduction of the forefoot, assessment from multiple planes using three-dimensional (3D) US has advantages over 2D US should be performed. In obstetrical ultrasound, the utilization of 3D US augments diagnostic capacities for congenital heart disease, musculoskeletal disease, and central nervous system disease, thus improving diagnostic accuracy. Images displayed using three simultaneous orthogonal planes and/or rendered images facilitate the assessment of structures from various angles, while also enabling the measurement of their volume [19]. Milan et al. first assessed the suspected clubfoot based on Surface-mode reconstruction of the lower limb by using 3D US [20]. They reported that 3D US had a higher PPV (89.9% vs. 67.8%) for diagnosing clubfoot than 2D US. 3D ultrasound presents a straightforward and convenient approach to assessing the fetus, as it is less constrained by fetal position or movement. It not only minimizes operator effort but also enables volumetric measurement and complete limb structure reconstruction. Consequently, this technique has been proposed as beneficial in diagnosing various musculoskeletal conditions.

Building on this background, we reconstructed the three-dimensional structure of the lower limb of a fetus using 3D US. We measured deviations in equinus, varus, calcaneopedal block, and forefoot adduction, along with calf volumes and calf circumferences, which are employed in postnatal lesion evaluation. Our objective is to identify ultrasound factors that can predict the need for clubfoot treatment.

## 2. Materials and Methods

### 2.1. Study Population

Identified patients were those who were suspected of fetal clubfoot and received prenatal care at our institution (Chung-Ang University Hospital, Seoul, Republic of Korea) and postnatal neonatal evaluation between 1 January 2018, and 31 August 2021. The inclusion criterion for this study was singleton pregnancy with prenatally suspected isolated clubfoot. Intrauterine death, neonatal death, neonate with complex malformations, and multiple pregnancies were excluded. Finally, 31 of the 39 pregnancies were examined. Among the 8 cases that were excluded, 5 presented with multiple anomalies, 1 resulted in neonatal death, and 2 were twin pregnancies.

### 2.2. Demographic Characteristics

The maternal and neonatal demographic data and medical history including maternal age, body mass index, preeclampsia, gestational diabetes, delivery mode, gestational age of delivery, preterm delivery, neonatal sex, birth weight, and postnatal treatment were obtained from medical records conducted by healthcare providers. We collected data on mechanical factors fetal breech presentation and maternal uterine abnormalities such as uterus didelphys, arcuate uterus, bicornuate uterus, and septate uterus. The constriction of the uterine cavity and fetal presentation can restrict fetal foot movement, resulting in clubfoot due to reduced joint and/or bone formation. Postnatal treatment was classified into two types: Ponseti conservative treatment, which includes serial cast treatment and Achilles tendon percutaneous tenotomy, and open surgery, which involves posterior or posteromedial release. The observation period was defined as the second year of life, and pregnant women who gave birth in August 2023 were the last subjects.

### 2.3. Sonographic Measurements

One skilled operator reviewed and assessed all the US recordings on the day of measurement. Following the as low as reasonably achievable principle, the recordings were recalled and measured after the US examination to reduce the time of exposure. For all US examinations, the 3D mode of Voluson™ E10 (© 2021 General Electric Company, New York, NY, USA) model was used. All patients had a detailed ultrasound examination, including a fetal echo, at the time of the first visit for the detection of associated anomalies with clubfoot.

In the coronal plane, where both the tibia and fibula are visible via 2D US, the cross-section of a normal foot is not visible (Figure 1A). In cases where clubfoot is suspected, the lateral surface of the affected foot is observed simultaneously (Figure 1B). Following the saving of a 2D image indicating clubfoot, 3D ultrasound is employed to perform evaluations.

We were able to freely handle the 3D captured image at 360 degrees and reproduce the desired planes, reconstructing the sagittal, coronal (anterior and posterior), and horizontal planes. Below is an explanation of the deviation value measured at each plane (Figure 2). 1. Equinus deviation: the fixation of the foot in a plantar inward position in the sagittal plane. We measured the deviated angle between the line perpendicular to the tibia and the midline of the dorsum. 2. Varus deviation: the medial deviation of the foot in the posterior coronal plane. We measured the angle between the line parallel to the tibia and the midline of the calcaneus 3. Calcaneopedal block deviation: derotation of the calcaneal forefoot block in the frontal coronal plane. We measured the angle between the medial line of the calf and the inner line of the talus. 4. Forefoot adduction in the horizontal plane. We measured the angle between the midline of the plantar and the inner line of the big toe. Calf volume and circumference were also measured because calf muscle dystrophy is a significant clinical feature of congenital clubfoot and is one of the scoring factors in the Demeglio method. First, we created a plane in which the tibia and fibula are observed. Second, designating the end of the ossified tibia bone, we divided it horizontally into five sections and measured the circumference using calipers on the skin surface. The largest calf circumference and automatically calculated calf volume were recorded (Figure 3).

### 2.4. The Postnatal Assessment and Treatment

After birth, an orthopedic surgeon assessed the clubfoot using the Dimeglio and Pirani scores and determined the therapeutic decision. The Ponseti method is a conventional treatment, which involves the placement of a series of five to seven casts over a few weeks or months. The first cast was placed on the foot within 48 h of birth to improve treatment efficacy. When the patient required further correction of the feet, an Achilles tendon percutaneous tenotomy was performed.

### 2.5. Statistical Analysis

The Mann–Whitney U test was used to compare continuous variables, and chi-square or Fisher’s exact test was used to compare categorical variables. The chi-squared test was used to estimate categorical parameters. The Spearman correlation coefficient was used to estimate the correlation between prenatal and postnatal parameters. ROC analysis was conducted to compare the performance of prediction models. The comparison was based on the AUC values of the models. The optimal cutoff values were determined as the point which maximizes the sum of the sensitivity and specificity. The sensitivity, specificity, positive predictive value (PPV), and negative predictive value (NPV) were also calculated to assess the diagnostic utility of each parameter in predicting the postnatal treatment of clubfoot. Statistical analyses were performed using R software (version 3.2.4; R Foundation for Statistical Computing, Vienna, Austria), and statistical significance was accepted where *p* values were <0.05.

## 3. Results

Of the 31 pregnancies 47 feet, 16 cases of bilateral clubfoot (32 feet), and 15 cases (15 feet) of unilateral clubfoot were suspected. The median gestational weeks when the ultrasound was performed was 26 weeks of gestation. The median age at delivery was 38 + 4 weeks. There was no difference in fetal presentation between the two groups, including vertex, breech, or transverse presentation. Uterine structural abnormalities affecting fetal presentation were not present in any of our subjects. Of neonates, 17 (54.8%) were male and 14 (45.2%) were female. After delivery, 78.7% (37/47) were treated and 21.2% (10/47) were observed without treatment. Additional open surgery (posterior or posteromedial release) was not performed in all patients. Table 1. presents the results of the comparison of baseline characteristics between the treated and non-treated groups, indicating that no statistically significant differences were observed.

### 3.1. The Differences in Deviated Angle, Calf Volume, and Calf Circumference between Treated Group and Non-Treated Group

Table 2 presents a comparison of deviated angles measured in the relative plane using 3D US between the two groups. The degree of hindfoot varus deviation was greater in the treated group than in the non-treated group (60.5° [50.6–71.4] vs. 46.6° [25.8–52.2], *p* = 0.042). The deviation of calcaneopedal block in the treated showed a higher degree than the non-treated group (65.6° [58.2–88.2] vs. 26.6° [23.0–32.8], *p* < 0.05). No significant differences were identified in equinus and forefoot adduction. The volume and circumference of the calf had no differences between the two groups. When assessing clubfoot through the conventional 2D US measurement, no statistically significant variance in the angle between the lower leg and foot was observed (80° [66–100] vs. 95° [61–103.8], *p* = 0.794).

### 3.2. The ROC Curve for Prediction of Postnatal Treatment

In the ROC curve analysis (Figure 4) for the prediction of postnatal treatment, the varus angle demonstrated an area under the curve (AUC) of 0.73 and suggested a diagnostic performance with 73% sensitivity, 80% specificity, 93% PPV, and 44% NPV at a diagnostic threshold of 53.4 degrees. The AUC of calcaneopedal block deviation showed a higher predictable value with 0.98 including 97% sensitivity, 90% specificity, 90% PPV, and 97% NPV at a diagnostic threshold of 46.2 degrees than the AUC of varus deviation (Table 3). To enhance the accuracy of the prediction, we conducted a ROC curve analysis by incorporating varus deviation and calcaneopedal block deviation values. An AUC of 0.96 represented with 81% sensitivity, 100% specificity, 100% PPV, and 59% NPV at a threshold of 111.8 degrees. 

### 3.3. Subgroup Analysis

#### 3.3.1. The Correlation between the Number of Casts and the Degree of Deviation

We performed a sub-analysis on the number of casts administered in the treated group and correlations with the level of deviation, calf volume, and calf circumference by Spearman correlation. No significant linear relationship was found between any variable and cast count (no figures are provided).

#### 3.3.2. Differences in Calf Volume and Calf Circumference Values between Two Calves with Unilateral and Bilateral Clubfoot

In cases of unilateral clubfoot (Figure 5A), as based on data obtained from 10 pregnancies (7 treated groups and 3 untreated groups), excluding 1 case with inadequate ultrasound data, no significant differences were observed with regards to calf volume (17.3% [12.3–24.8] vs. 18.2% [13.8–25.9], *p* = 0.938) and calf circumference (4.2% [1.5–8.9] vs. 20.0% [15.1–24.9], *p* = 0.114) between treated and the non-treated groups. Of the 18 pregnancies who were suspected to have bilateral clubfoot, 5 patients (10 feet) were excluded due to insufficient ultrasound data. Thus, 13 patients (26 feet) were available to compare the discordance in the treated (10 patients) and untreated (3 patients) groups. (Figure 5B). There were no differences in calf volume discordancy (10.3% [5.8–11.9] vs. 2.2% [1.3–12.5], *p* = 0.398) and calf circumference discordancy (3.0% [2.6–9.2] vs. 1.4% [1.2–14.9], *p* = 0.612). 

## 4. Discussion

Notably, our study identified ultrasound-specific factors that predict postnatal treatment for suspected clubfoot based on surface-mode reconstruction images using 3D ultrasound. We applied the dimeglio method, a physical examination method used to determine treatment options after birth, to images that reproduce the three-dimensional structure of the lower limb. A previous study reported the use of surface-mode reconstruction images with 3D ultrasound for suspected clubfoot, but this approach was considered imprecise because it relied on overall image evaluation to improve the diagnosis rate without establishing a correlation with specific ultrasound factors [20]. Our data suggested that the degree of varus deviation and calcaneopedal block was higher in the treated group than in the non-treated group. The cutoff value calcaneopedal block deviation indicated an AUC of 0.98 which is considered to be within an acceptable range and high accuracy with 96%. Prenatal diagnosis of suspected clubfoot was performed using 2D US, which is a typical method of observing the deviating angle between the calf and foot in the coronal plane. Although the postnatal Dimeglio scoring system includes four parameters (equinus deviation, varus deviation, derotation of the calcaneo-forefoot block, and forefoot adduction), and four gravity signs (plantar crease, medial crease, cavus retraction, and fibrous musculature), 2D US prenatal assessment has considered only one parameter and is insufficient for assessing the severity and postnatal outcomes of congenital foot conditions [21]. A few previous studies have attempted to evaluate severity and predict postnatal outcomes using 2D US. Glotzbecker, M.P. et al. developed a sonographic scoring system that classifies the severity of clubfoot into three categories (mild, moderate, and severe) and demonstrated diagnosed severe cases based on the scoring system, leading to a decrease in the FPR (7–19%) [16,22]. Following this result, a prospective study based on the aforementioned scoring system suggested that the accuracy of diagnosis in cases with severe, moderate, and mild scores was 94%, 70%, and 25%, respectively [22]. However, this study did not provide a measurement tool for improving the diagnostic accuracy of mild clubfoot.

Recently, Lanna et al. attempted to improve prenatal diagnosis using 2D US with a multi-planar view. The angle between the long axis of the foot and lower leg; foot width, and length, the width-to-length ratio of the foot; tibia length, and calf width ratio were examined to predict postnatal treatment. They suggested that the angle between the long axis of the foot showed the highest accuracy at a diagnostic threshold of 80 degrees (AUC 0.98, confidence interval 0.94–1.00) [18]. We also recorded the angles using conventional 2D US to evaluate the diagnostic superiority of the angle using 3D US but did not obtain significant difference values between the treated and non-treated groups. This result is inconsistent with the study by Lanna et al., which found that the angle in the treatment group was higher than that in the non-treatment group in 2D US.

We hypothesized that the treated group would exhibit a reduction in calf volume and circumference in utero due to calf muscle dystrophy [23]. Calf volume and circumference exhibited no significant variance between the treated and untreated groups. The nomogram from previous research [24] indicates that calf muscle size and diameter increase with each week of gestation. The different weeks of gestation at which ultrasound was performed for each subject in our study may have contributed to these results. We aimed to compare the calf development of our subjects with normal calf development based on gestational weeks, using the calf nomogram developed by R. Hershkovitz et al. However, the method for measuring calf development recommended in the previous nomogram was unsuitable, given that our study applied a different assessment method through the use of 3D US.

Additionally, we hypothesize that unilateral pathologic clubfoot results in decreased calf volume and circumference in the affected foot compared to the contralateral normal foot. In contrast, bilateral pathologic clubfoot does not show any differences in either factor. We performed a subgroup analysis to compare calf volume and circumference discordance between the treated and non-treated groups in unilateral clubfoot and bilateral clubfoot, respectively. In bilateral clubfoot, the volume and circumference of both calves did not differ between the treatment and non-treatment groups, but in unilateral clubfoot, there was also no difference between the treatment and non-treatment groups. This is inconsistent with our hypothesis. We consider that the number of subjects used in this analysis was small, 10 (unilateral clubfoot) and 13 (bilateral clubfoot), and therefore not statistically significant. To overcome this limitation, we are conducting a further study to establish nomograms for normal fetal calf volume and circumference using 3D US.

The importance of calcaneopedal block derotation as a determinant of treatment outcomes in orthopedics is widely recognized [25,26]. Brazell et al. reported that correction of the equinus deformity should only be considered after reaching a calcaneopedal block derotation of at least 20–30 degrees [25]. There are limitations in accessing 2D US for assessing calcaneopedal block derotation in utero. To assess calcaneopedal block derotation, measure the angle after creating a frontal plane with the fetus’ knee facing forward. In the 2D method, the angle is measured in a plane where both the tibia and fibula are visible, but this is not necessarily the exact coronal plane. We utilized 3D US to establish a precise coronal plane that can assess calcaneopedal block deviation and ultimately overcome clinical limitations. Our findings suggested that this parameter is the most accurate predictor in prenatal ultrasound.

Clubfoot is classified as a type and complex type, depending on the presence of associated deformity. Isolated clubfoot has been reported to have chromosomal abnormalities in 3% [27], whereas complex clubfoot has been reported in 20% [28]. For this reason, detailed ultrasonography and fetal echocardiography are recommended for the initial management of suspected clubfoot. In particular, it is important to perform both intrauterine precision ultrasound and fetal echocardiography when considering the incidence of chromosomal abnormalities such as trisomy 18 or 21 associated with congenital heart disease with isolated clubfoot at 5–22% [4,29,30]. A previous study reported that prenatal diagnosis of isolated clubfoot was associated with postnatal diagnosis of additional malformations in more than half of the cases [31]. With the advancement of ultrasound technology, it has been reported that only 5% of patients diagnosed with isolated clubfoot prenatally have additional deformities discovered after birth [28].

Our study has several limitations: Firstly, we didn’t examine the verification of the intraobserver and interobserver reliability of our measurement method. Further assessment of intra- and interobserver reliability and validation of the cut-off value for calcaneopedal block to predict clubfoot is required for actual clinical application in prenatal suspected clubfoot. Second, our retrospective study included only patients with isolated clubfoot, excluding cases in which associated anomalies were diagnosed prenatally or postnatally, and we could not provide the information. Thirdly, the evaluation of the calf muscles and calf circumference, which are important factors in clubfoot treatment, was inadequate. Therefore, we could not build a predictive model combining the dislocation angle in each plane, calf circumference, and calf volume as planned at the beginning of the study.

Our 3D US assessment tool for predicting postnatal clubfoot outcomes suggested that a multiplane approach could improve prenatal diagnostic accuracy. This improved assessment and prediction rate can provide accurate information and decrease the emotional burden of parental counseling. This study needs more prospective data including calf volume and circumference to evaluate muscular dystrophy associated with severe clubfoot.

## 5. Conclusions

Our study suggested a new measure of calcaneopedal block deviation by US assessment to replace the conventional 2D US evaluation method for prenatal clubfoot diagnosis. The ankle dislocation angle measured by conventional 2D US did not show a significantly higher value in the treated group compared to the non-treated group. In contrast, the calcaneopedal block angle measured by 3D US showed a higher value in the treated group than in the non-treated group, with an area of 46 degrees as the standard, which showed a high value of 0.98 area under the curve. Calcaneopedal block deviation holds significant importance for postnatal clubfoot assessment, and our findings demonstrate that it can be a reliable indicator of clubfoot in the fetus’s prenatal 3D ultrasound assessment. Prenatal measurement of calcaneopedal block deviation could enhance the accuracy of diagnosing clubfoot during prenatal care, which was earlier assessed solely through 2D ultrasound. Furthermore, these findings underscore the necessity of implementing 3D ultrasound in the diagnosis of congenital musculoskeletal disorders, including clubfoot. 

## Figures and Tables

**Figure 1 diagnostics-14-00117-f001:**
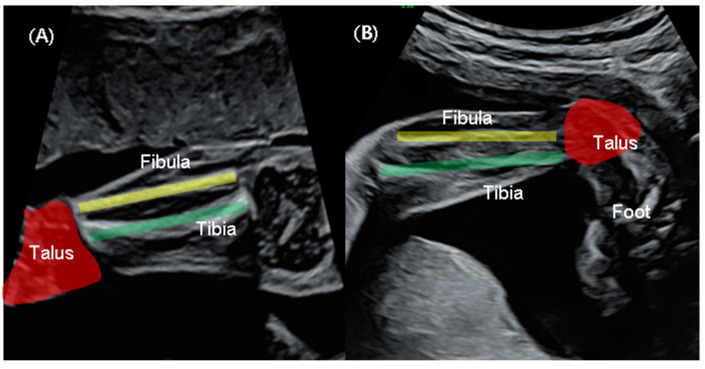
Two-dimensional sonographic view (**A**) Normal fetus: the tibia (green line) and the fibula (yellow line) are observed in the frontal coronal plane as talus (red colored), but the foot is not observed. (**B**) Suspected clubfoot: the tibia and fibula are observed in the coronal plane as the lateral aspect of the foot, with the foot extended and inverted.

**Figure 2 diagnostics-14-00117-f002:**
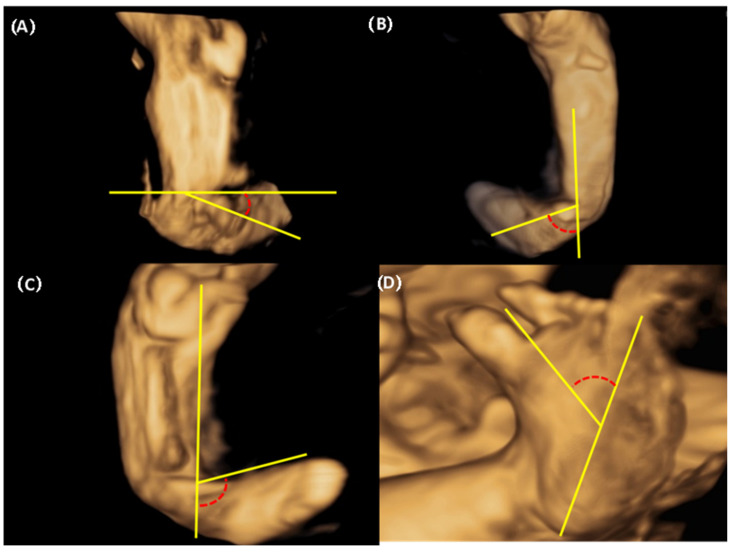
The three-dimensional assessment of deviation between foot and calf and plantar deviation of the foot (**A**) Equinus deviation in the sagittal plane (**B**) Varus deviation in the frontal plane (**C**) Derotation around the talus calcaneo-forefoot in the coronal plane (**D**) Adduction forefoot on forefoot in the horizontal plane.

**Figure 3 diagnostics-14-00117-f003:**
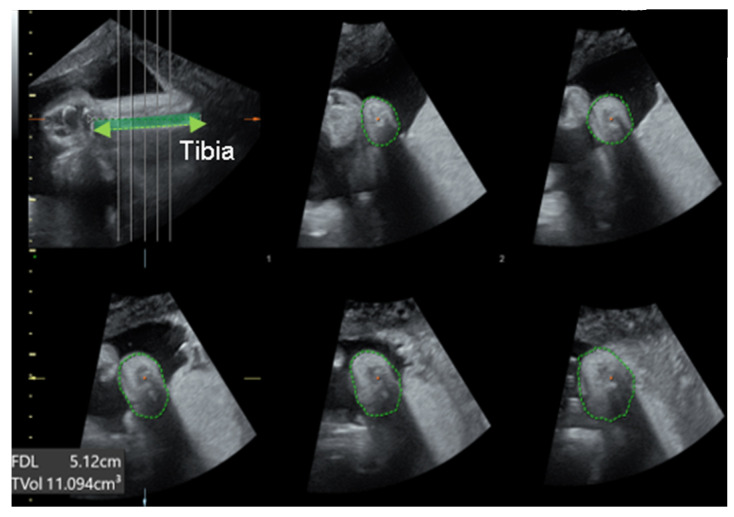
The measurement of calf volume and circumference using 3D US. The green dotted line shown in the top left picture indicates tibia. The green circular dotted line shown in the remaining five pictures indicates the calf circumference.

**Figure 4 diagnostics-14-00117-f004:**
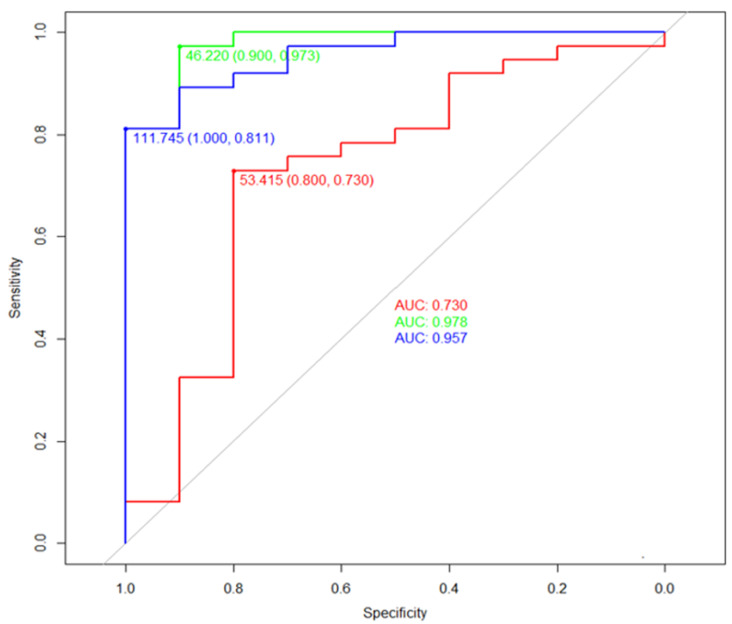
The diagnostic thresholds of angle provided by ROC curve analysis for prediction of congenital clubfoot using 3D USG. The green line demonstrated the ROC curve of the calcaneopedal block angle. The red line demonstrates the ROC curve of the varus angle. The blue line demonstrated the ROC curve of the sum of the varus and calcaneopedal block angle. ROC curve, receiver operating characteristic curve; 3D, three-dimensional; US, ultrasonography; AUC, the area under the curve.

**Figure 5 diagnostics-14-00117-f005:**
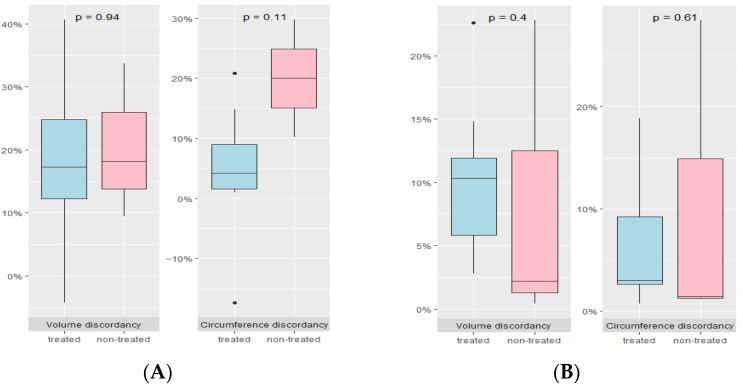
Discordancy in calf volume and calf circumference between the treated and non-treated groups. (**A**) Unilateral clubfoot (**B**) Bilateral clubfoot.

**Table 1 diagnostics-14-00117-t001:** Baseline characteristics of the treated group and non-treated group.

	Treated Group (N = 37)	Non-Treated Group (N = 10)	*p*
Maternal age (years)	32 (31–35)	34 (30.5–35.7)	0.514
BMI (m^2^/kg)	27.05 (24.5–28.9)	28.7(26.9–9.2)	0.475
Preeclampsia	1 (2.7%)	1 (10%)	0.895
Gestational diabetes	0	0	NA
GA of US evaluation (weeks)	25 +6 (23 + 1–27 + 3)	26 + 4 (25 + 4–30 + 2)	0.088
Presentation before delivery			0.407
Cephalic	35 (94.6%)	8 (80%)	
Breech	2 (5.4%)	2 (20%)	
Transverse	0	0	
Cesarean delivery	35 (67.6%)	3 (30%)	0.074
GA of delivery (weeks)	38 + 2 (36 + 5–39 + 3)	38 + 2 (36 + 3–39.0)	0.865
Preterm delivery (<37 weeks)	1 (2.7%)	1 (10%)	0.895
Neonatal sex (male)	21 (56.8%)	3 (30%)	0.252
Birth weight (g)	3120 (2810–3531)	3037.5 (2978–3233)	0.474
Unilateral clubfoot	12 (32.4%)	3 (30%)	1.000
Serial cast treatment	37	0	<0.05
Achilles tendon percutaneous tenotomy	22	0	<0.05

Median (interquartile range) or number (percentage). BMI; body mass index, US; ultrasonography, GA; Gestational age.

**Table 2 diagnostics-14-00117-t002:** The differences in sonographic variables between the treated group and the non-treated group.

	Treated Group (N = 37)	Non-Treated Group (N = 10)	*p*
3D US variables			
Equinus (°)	12.1 (3.3–22.5)	15.3 (6.4–22.0)	0.612
Varus (°)	60.5 (50.6–71.4)	46.6 (25.8–52.2)	0.026
Derotation of CPB (°)	67.6 (58.2–88.2)	26.6 (23.0–32.8)	<0.05
Adduction of the forefoot (°)	22.4 (15.9–30)	19.7 (17.3–25.0)	0.649
Volume of calf (cm^3^) ^a^	5.16 (3.57–8.68)	6.59 (5.7–8.18)	0.257
Circumference of calf (cm) ^a^	6.5 (5.79–8.03)	7.68 (7–8.8)	0.073
2D US variable			
The angle between the long-axis of the foot and the lower leg (°)	80 (66–100)	95 (61–103.75)	0.794

Median (interquartile range) or number (percentage). CPB: calcaneopedal block, ^a^ Calf volume and calf circumference were utilized for 32 cases in the treated group and 9 cases in the non-treated group due to insufficient ultrasound data in 6 cases.

**Table 3 diagnostics-14-00117-t003:** Accuracy, sensitivity, specificity, positive predictive value, and negative predictive value of derotation of calcaneopedal block.

Threshold	Sensitivity	Specificity	NPV	PPV	Accuracy	AUC
46°	0.97	0.90	0.90	0.97	0.96	0.98

NPV: negative predictive value, PPV: positive predictive value, AUC: area under curve.

## Data Availability

The data that support the findings of this study are available from the corresponding author upon reasonable request.

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
