# Peer review of "Improving Prenatal Diagnosis Precision for Congenital Clubfoot by Using Three-Dimensional Ultrasonography"

_diagnostics, 2024, doi:10.3390/diagnostics14010117_

Round 1
Reviewer 1 Report
Comments and Suggestions for Authors
Dear authors
clubfoot is among the most common fetal medicine findings
therefore congratulations your paper is of interest the study well conducted and immeges really nice and useful
i have minor revisions to suggest
please remind readers that in front of any fetal abnormality according to the most relevant international guidelines it must me suggested to offer a fetal echocardiography
(read and cite
PMID: 36786908
DOI: 10.1002/uog.26224 )
please add a simple and clear table with major findings
and tips for operators to help with the diagnosis according to your findings
best regards
Author Response
Response: We thank for the reviewer’s valuable comment. To address the request of Reviewer, we have amended the sentence as following.
(line 334 - 341)
“Clubfoot is classified as type and complex type, depending on the presence of associated deformity. Isolated clubfoot has been reported to have chromosomal abnormalities in 3%, whereas complex clubfoot has been reported in 20%. For this reason, detailed ultrasonography and fetal echocardiography are recommended for the initial management of suspected clubfoot. In particular, it is important to perform both intrauterine precision ultrasound and fetal echocardiography when considering the incidence of chromosomal abnormalities such as trisomy 18 or 21 associated with con-genital heart disease with isolated clubfoot at 5-22%. A previous study re-ported that prenatal diagnosis of isolated clubfoot was associated with postnatal diagnosis of additional malformations in more than half of the cases. With the advancement of ultrasound technology, it has been reported that only 5% of patients diagnosed with isolated clubfoot prenatally have additional deformities discovered after birth.”
Response: To help readers better understand the clinical application of our results, we added a simple and clear table with major findings to manuscript (Table 3). Please see the attachment

Reviewer 2 Report
Comments and Suggestions for Authors
Your article is about clubfoot prenatal diagnosis and correlation with postpartum treatment. Subject is interesting for healthcare providers for antepartum counselling as well as for future parents to understand better the pathology therefore the conclusions are very important.
However, there are some issues addressed below:
- clubfoot is also associated with uterine malformations and fetal lie/presentation - these informations should also be included in demografic/sonographic section of your target group.
- in discussion section you mentioned your study as the first one in this section - please see "Limb deformities and three-dimensional ultrasound Limb deformities and three-dimensional ultrasound - Milan Kos J Perinat Med 2002;30(1):40-7. doi: 10.1515/JPM.2002.006" and rephrase accordingly
- your main conclusion is that postnatal prediction is better while assessing the calcaneopedal block deviation in 3D however if you compare prediction of standard 2d with 3d assesment you should include this in your conclusion.
Author Response
Comments and Suggestions for Authors
- clubfoot is also associated with uterine malformations and fetal lie/presentation - these informations should also be included in demografic/sonographic section of your target group.
Response: We thank for the reviewer’s valuable comment. To address the request of Reviewer, we have amended the sentence as following at 2.2. Demographic characteristics and 3. Results.
(line 115-119)
“We collected data on mechanical factors fetal breech presentation and maternal uterine abnormalities such as uterus didelphys, arcuate uterus, bicornuate uterus, and septate uterus. The constriction of the uterine cavity and fetal presentation can restrict fetal foot movement, resulting in clubfoot due to reduced joint and/or bone formation.”
(line 195- 197)
“There was no difference in fetal presentation between the two groups, including vertex, breech, or transverse presentation. Uterine structural abnormalities affecting fetal presentation were not present in any of our subjects”
- in discussion section you mentioned your study as the first one in this section - please see "Limb deformities and three-dimensional ultrasound Limb deformities and three-dimensional ultrasound - Milan Kos J Perinat Med 2002;30(1):40-7. doi: 10.1515/JPM.2002.006" and rephrase accordingly
Response: We thank the reviewer for the useful advice. To avoid confusion, we revised the content to state that this is the first study to identify ultrasound parameters that predict clubfoot using 3D ultrasound. Furthermore, we added a study on clubfoot evaluation using 3D ultrasound to the paper mentioned by the reviewer.
(line 262-269)
“Notably, our study identified ultrasound-specific factors that predict postnatal treatment for suspected clubfoot based on surface-mode reconstruction images using 3D ultrasound. We applied the dimeglio method, a physical examination method used to determine treatment options after birth, to images that reproduce the three-dimensional structure of the lower limb. A previous study reported the use of surface-mode reconstruction images with 3D ultrasound for suspected clubfoot, but this approach was considered imprecise because it relied on overall image evaluation to improve the diagnosis rate without establishing a correlation with specific ultrasound factors.”
- your main conclusion is that postnatal prediction is better while assessing the calcaneopedal block deviation in 3D however if you compare prediction of standard 2d with 3d assessment you should include this in your conclusion.
Response: We thank for the reviewer’s valuable comment. As you mentioned, we add the above to the conclusion.
(line 366-370)
“The ankle dislocation angle measured by conventional 2D US did not show a significantly higher value in the treated group compared to the non-treated group. In contrast, the calcaneopedal block angle measured by 3D US showed a higher value in the treated group than in the non-treated group, with an area of 46 degrees as the standard, which showed a high value of 0.98 under the curve.”
Reviewer 3 Report
Comments and Suggestions for Authors
We would like to thank the author team for accumulating so many cases and accumulating valuable experience for prenatal diagnosis using 3D ultrasound. The article is fluent in writing and has great significance for clinical guidance.
Author Response
We appreciate the positive comments of the reviewer.
Reviewer 4 Report
Comments and Suggestions for Authors
The manuscript "Improving prenatal diagnosis precision for congenital clubfoot by using three dimensional ultrasonography" is an interesting manuscript about the role of 3D ultrasound in the prenatal evaluation of congenital clubfoot. The work is complete and well structured, giving to clinicians and scientific literature important lessons. The design of the project is appropriate and the images are satisfactory. The language is acceptable. The description of the limitations of the study is confusing, it looks like the article is not complete and it is only a preliminary work of future studies: I suggest improving this part and including the strengths of the work. Otherwise, it represents a valid work and it allows focusing attention on possible future studies on prenatal diagnosis of clubfoot
Author Response
Response: We appreciate the positive comments of the reviewer. As previously stated, this study is limited due to its preliminary study design, small number of subjects, and unproven reproducibility. To avoid confusion, we plan to collect more data in the future and present additional high-quality research results that have undergone appropriate validation processes.